# Variants in *GSTZ1* Gene Underlying Maleylacetoacetate Isomerase Deficiency: Characterization of Two New Individuals and Literature Review

**DOI:** 10.3390/genes16091009

**Published:** 2025-08-26

**Authors:** Ferdinando Barretta, Fabiana Uomo, Alessandra Verde, Mariagrazia Fisco, Giovanna Gallo, Lucia Albano, Daniela Crisci, Cristina Mazzaccara, Pietro Strisciuglio, Margherita Ruoppolo, Simona Fecarotta, Giancarlo Parenti, Giulia Frisso, Alessandro Rossi

**Affiliations:** 1CEINGE Biotecnologie Avanzate–Franco Salvatore s.c.ar.l., 80131 Naples, Italy; barretta@ceinge.unina.it (F.B.); uomo@ceinge.unina.it (F.U.); fisco@ceinge.unina.it (M.F.); gallo@ceinge.unina.it (G.G.); albano@ceinge.unina.it (L.A.); crisci@ceinge.unina.it (D.C.); cristina.mazzaccara@unina.it (C.M.); mruoppol@unina.it (M.R.); 2Department of Advanced Biomedical Science, University of Naples “Federico II”, 80131 Naples, Italy; 3Department of Translational Medical Science, Section of Paediatrics, University of Naples “Federico II”, 80131 Naples, Italy; verde-alessandra@libero.it (A.V.); pietro.strisciuglio@unina.it (P.S.); parenti@unina.it (G.P.); alessandro.rossi@unina.it (A.R.); 4Department of Molecular Medicine and Medical Biotechnology, University of Naples “Federico II”, 80131 Naples, Italy; 5Department of Integrated Maternal-Fetal Care, “Federico II” University Hospital, 80131 Naples, Italy; simona.fecarotta@unina.it

**Keywords:** tyrosinemia type 1, newborn screening, succinylacetone, maleylacetoacetate isomerase deficiency, *GSTZ1* mutations, mRNA analysis

## Abstract

Introduction: Elevated succinylacetone (SA) is the hallmark of tyrosinemia type 1, which requires immediate treatment. Mild SA elevation has also been recently reported in maleylacetoacetate isomerase deficiency (MAAID). Methods: We report on two cases of MAAID, review clinical features of MAAID and discuss its management. Results: Both cases displayed elevated SA and normal Tyrosine levels at newborn screening. Case 1 showed intermittent SA elevation; Nitisinone and dietary treatment were started, then discontinued after the identification of two variants in the *GSTZ1* gene and the definitive diagnosis of MAAID. Case 2, showing no SA elevation at the confirmatory tests and two variants in the *GSTZ1* gene, did not start treatment. mRNA analysis confirmed the pathogenicity of the c.68-12G>A variant, found in both patients. Discussion: MAAID should be considered in newborns showing elevated SA and no variants in the *FAH* gene. Our study reports for the first time the course of SA in a patient affected by MAAID. Furthermore, it expands the molecular epidemiology of this rare disease, also investigating the pathogenicity of a novel splicing mutation. Although our data argue against medical treatment in MAAID, longer follow-up data are warranted.

## 1. Introduction

Hypertyrosinemia is a biomarker of several inherited metabolic disorders due to a defect in the enzymes involved in tyrosine metabolism, including hereditary Tyrosinemia type I (HT-1), type II (HT-2), type III (HT-3), Alkaptonuria, and Hawkinsinuria. HT-1 is caused by biallelic variants in the fumarylacetoacetate hydrolase (FAH) gene and can present with acute hepatic and renal failure within the first weeks of life [1]. Nitisinone (NTBC) and tyrosine/phenylanine-restricted diet significantly decreases the acute complications of HT-1 and should be started as soon as possible [2]. Hence, HT-1 is included in newborn screening (NBS) programs, enabling early diagnosis and treatment for affected individuals [3]. Elevated succinylacetone (SA) on a dried blood spot (DBS) has long been considered pathognomonic of HT-1 [4]. However, two additional conditions associated with DBS-SA elevation have been identified, namely partial FAH deficiency and maleylacetoacetate isomerase (MAAI) deficiency (MAAID) [5]. The *GSTZ1* gene encodes maleylacetoacetate isomerase, a bifunctional enzyme that plays a critical role in both tyrosine catabolism and detoxification of xenobiotic compounds, such as dichloroacetate. Within the tyrosine degradation pathway, the enzyme catalyzes the isomerization of maleylacetoacetate (MAA) to fumarylacetoacetate (FAA), a step immediately upstream of fumarylacetoacetate hydrolase. Deficiency of this enzyme leads to the accumulation of upstream intermediates, including succinylacetone, a toxic metabolite traditionally regarded as pathognomonic for tyrosinemia type I. However, recent evidence has identified biallelic variants in the *GSTZ1* gene as the molecular cause of maleylacetoacetate isomerase deficiency, a rare autosomal recessive condition that can mimic HT-1 in newborn screening due to mild or intermittent elevations of SA. Despite the biochemical overlap, MAAID appears to have a markedly milder clinical course than classical HT-1, and follow-up of reported patients has not revealed liver dysfunction or neurological complications. Given the expanding implementation of succinylacetone-based newborn screening and the increasing use of genomic sequencing for diagnostic confirmation, understanding the clinical implications of *GSTZ1* variants is crucial for distinguishing true HT-1 from benign or less severe metabolic variants. This is particularly important to prevent unnecessary treatment with nitisinone and to reduce the psychological and logistical burden associated with false-positive results. Less than 20 people with MAAID have been reported, with clinical follow-up only available for 9 of them; among these, bi-allelic variants in the *GSTZ1* gene were identified in only eight individuals [6,7]. To date, the clinical relevance and management of MAAID as well as the long-term prognosis remain unclear, warranting additional data.

We report on two newborn cases displaying SA elevation at NBS and no variants in the FAH gene diagnosed as MAAID following genetic testing, revealing the presence of biallelic variants in the *GSTZ1* gene. Interestingly, for one of the two cases, we describe the course of SA throughout the follow-up period to assess the effect of treatment. Characterization of these two new cases with MAAID expands the knowledge base of this rare condition and contributes to the optimization of the management plan for newborns recalled at NBS for elevated SA.

Finally, we provide a review of currently available evidence on this condition.

## 2. Materials and Methods

### 2.1. Newborn Screening

Neonatal metabolic screening and subsequent genetic testing were conducted at CEINGE—Biotecnologie Avanzate Franco Salvatore (Naples, Italy), the designated Regional Center for Newborn Screening in Campania and a reference facility of the National Health Service for Clinical Molecular Biology and Laboratory Genetics. Blood samples from newborns were obtained via heel prick and applied to Schleicher & Schuell 903-grade filter paper cards (Whatman, Dassel, Germany). According to clinical guidelines, sample collection for screening is recommended within 48 to 72 h after birth. Dried blood spot (DBS) cards were delivered daily to the laboratory via courier service [8]. The analytical procedure included the extraction of amino acids and acylcarnitines, which serve as key biomarkers for the detection of inherited metabolic disorders covered by the national newborn screening program, as established by Italian legislation (2016–2017). The analysis was performed using liquid chromatography-tandem mass spectrometry (LC-MS/MS), as previously described [9]. Screening results were classified as “positive” when one or more biomarker concentrations exceeded or fell below established threshold values. These cutoffs are regularly re-evaluated by the laboratory, based on percentile distributions and data from false-positive cases. Whole blood and serum samples were stored at temperatures between 2 °C and 8 °C. Fresh urine samples were collected in sterile tubes and frozen for preservation. Quantification of serum amino acids was performed using high-performance liquid chromatography (HPLC), following the protocol described in a previous study [10].

### 2.2. Genetic Analysis and Bioinformatic Analysis

Case 1, collected before June 2020, was analysed by Sanger sequencing, using polymerase chain reaction amplification of all exons, the exon–intron boundaries and the 5′- and 3′-UTR regions of *FAH* and *GSTZ1* genes. Oligonucleotide sequences employed for amplification can be provided upon request.

In case 2, the molecular testing was carried out by analysis of a multigene NGS (Next Generation Sequencing) panel, including the five genes associated with hereditary tyrosinemia (i.e., *HPD*, *FAH*, *TAT*, *HGD*, and *GSTZ1*), according to a previously reported protocol [11,12]. In detail, library preparation was performed using the SureSelectXT-HS Target Enrichment System for Illumina Paired-End Multiplexed Sequencing Platforms protocol (Agilent Technologies, Santa Clara, CA, USA). Before proceeding to library preparation, the concentration and purity of the DNAs were assessed by the fluorimetric method, using Qubit dsDNA HS Assay Kit (Thermo Fisher Scientific, Waltham, MA, USA). The DNA’s integrity was evaluated by ScreenTape analysis on TapeStation (Agilent, Santa Clara, CA, USA). Genomic DNA was initially fragmented by a restriction enzyme. The resulting DNA fragments were enriched through hybridization with CCP17 probes (Custom Constitutional Panel; Design size: 17 Mb; Agilent Technologies, Santa Clara, CA, USA), followed by purification and PCR amplification to generate sequencing libraries. Each DNA sample was uniquely labeled with a barcode to enable multiplexing during the sequencing process. The final quality of the libraries was evaluated by the TapeStation system (Agilent Technologies, Santa Clara, CA, USA), which allows evaluation of both the library’s concentration and the size of the DNA fragments. The enriched and indexed libraries were sequenced on the NextSeq 550 platform (Illumina, San Diego, CA, USA). The reads alignment (BWA), the call (GATK), and the annotation (Annovar) of the variants were executed, respectively, using the Alissa Align & Call and Interpret (Agilent Technologies, Santa Clara, CA, USA). Variants were prioritized using a customizable analysis pipeline to identify pathogenic or potentially pathogenic mutations. Subsequently, we classified the variants according to American College of Medical Genetics and Genomics (ACMG) criteria [13]. Annotation and interpretation were supported by population frequency data and clinical databases, including ClinVar, HGMD, and VarSome. Furthermore, genetic variants detected in the patients were screened by Sanger sequencing, in their parents, to verify the family segregation.

We annotated identified variants according to *GSTZ1* transcript NM_145870.3, corresponding to protein: NP_665877.1

All procedures were in accordance with the standards of the Ethics Committees on human experimentation (Institutional and national) and with the Helsinki Declaration and were approved by the local Ethics Committee (N. 77/21).

### 2.3. Splice-Site Analysis

Alamut™ Visual Plus Software (version 1.10) was used to perform the in-silico analysis for the evaluation of the possible splicing variant (c.68-12G>A). To confirm the splicing alteration, we performed an in vitro splicing assay. The RNA was extracted from the peripheral blood of Patient 1 and his parents using the QIAamp RNA Blood Kit (Qiagen, Hilden, Germany). RT-PCR was performed as previously reported [14], using the iScript cDNA Synthesis Kit (Biorad, Hercules, CA, USA). The reaction mix was incubated in a thermal cycler under the following conditions: 5’ at 25 °C, 20’ at 46 °C and the final RT inactivation for 1’ at 95 °C. cDNAs were amplified by PCR Master Mix (Promega ™) using specific oligonucleotides (5’UTR_Forward TCACTGAGCCTTAGTCGTCG; EX6_Reverse TGCATCTCCTCTCCCACTTG; fragment amplification: 457 bp). Control reactions to detect the efficiency of cDNA synthesis were performed with specific primers (Forward TTGAAAGCCTCGTACCCTGG; Rev CACACCCACCAGATCCAAGA fragment amplification: 588 bp) for the human housekeeping *HMBS* gene (Hydroxymethylbilane Synthase). To verify the amplification, the reaction products were electrophoresed on 2% agarose gel and visualized through a UV transilluminator. Further amplified products were sequenced by the Sanger method.

## 3. Results

### Clinical Description and Molecular Analysis

Two newborns were referred to the Metabolic Disease Unit of our center due to increased SA, together with normal Tyrosine levels on NBS-DBS. The details of biochemical and clinical data are summarized in Table 1. Both displayed regular development at the last follow-up.

In Case 1, the confirmation biochemical testing could not rule out HT-1. Therefore, molecular testing for the FAH gene was ordered, and treatment with nitisinone (NTBC) (1.2 mg/kg/day) and a tyrosine/phenylalanine-restricted diet (tyrosine + phenylalanine 65 mg/kg/day, total protein 2.5 g/kg/day) was started. During this time, the boy displayed regular growth (Supplemental Appendix A) and no episodes consistent with tyrosinemic neurological crises. Since the molecular investigation of the *FAH* gene did not reveal any pathogenetic variants and considering the presence of maleyl-acetoacetic acid in the urine organic acid analysis, we performed *GSTZ1* sequencing. The c.68-12G>A and c.464_471delinsCTGGG (p.Val155_Asp157delinsAlaGly) variants were found. Therefore, at age 21 months, both NTBC and dietary treatment were discontinued. A modest increase in DBS- and UOA-SA were observed 11 months after treatment discontinuation (Figure 1), while tyrosine values, liver and kidney function tests resulted normal. At 4 years of age, the patient displayed regular development, obesity, and normal liver function (Table 1).

In Case 2, based on available data and building on previous experience from Case 1, no treatment was started, and the patient was included in a follow-up program (Table 1). Concurrently, genetic testing was requested using a multigene NGS panel. The genetic analysis highlighted the presence of the variants c.68-12G>A and c.295G>A, p.(Val99Met), in the *GSTZ1* gene. At 2 years of age, the girl displayed good clinical conditions, regular growth and development (Table 1).

Segregation analysis was conducted in both cases and demonstrated that the two variants identified in each patient in the *GSTZ1* gene are in *trans* (Supplemental Appendix A). Both patients shared the variant c.68-12G>A, which has already been described as associated with mild hyper-succinyl-acetonemia [6]. According to ACMG (American College of Medical Genetics) criteria [13], c.68-12G>A and c.464_471delinsCTGGG were classified as variants of uncertain significance (VUS); the c.295G>A variant was classified as likely pathogenic (LP).

The in silico analysis of the c.68-12G>A variant showed the possible formation of a cryptic splicing acceptor site in intron 2. To confirm this hypothesis, we carried out patient 1 RNA sequencing, which highlighted retention of 10 bp in the exon 2-3 junction, due to the creation of a cryptic splice site in position c.68-10 (Figure 2). Therefore, we were able to classify this variant as LP, including the PS3 criteria in the ACMG evaluation. Consequently, the c.464_471delinsCTGGG variant, being in trans with an LP variant (PM3 criterion in the ACMG evaluation), also became LP.

## 4. Discussion

SA has long been considered the pathognomonic marker of HT-1, allowing prompt diagnosis and management of affected individuals (1,2). The recent identification of SA elevation in MAAID has reshaped this paradigm and opened new challenges for healthcare professionals. MAAID is a rare metabolic disorder resulting from pathogenic variants in the *GSTZ1* gene. This enzyme catalyses the isomerization of maleylacetoacetate to fumarylacetoacetate in the tyrosine catabolism pathway, thus resulting in the accumulation of the toxic metabolite SA. However, SA levels appear to be lower and/or discontinuously elevated in MAAID as compared to HT-1 (7,8). Currently, clinical information is only available for eight molecularly confirmed MAAID individuals (Table 2).

While Berger et al. described one patient showing fatal hepatic and renal failure, for whom molecular confirmation was not available [15], in most of the cases, MAAID remains asymptomatic [8]. However, longitudinal follow-up has been described only in four cases, ranging from 10 to 32 years [8,9], and the long-term consequences of chronic/intermittent SA accumulation remain to be ascertained. Downregulation of GSTZ1 expression has been found in human hepatocellular carcinoma (HCC) tissues [16]. In GSTZ1-deficient HCC cells, SA has also been shown to increase HIF-1α and VEGF protein levels, thus promoting tumorigenesis [17]. These observations would argue in favour of an approach to decrease SA levels.

When NBS shows SA elevation, Yang et al. suggested starting NTBC only in the presence of liver dysfunction [6], which, from their experience, allows them to distinguish infants with HT1 from those with MAAID. In confirmed MAAID cases, Yang et al. suggested avoiding protein supplements and some medications, such as acetaminophen, dichloroacetate (DCA) and chloralhydrate [6]. However, the most appropriate management approach remains to be established. Additional cohort data, as well as longer follow-up of individuals with MAAID, are warranted.

In this study, we presented clinical, biochemical, and genetic characterization of two additional infants who showed SA elevation on NBS and were molecularly diagnosed with MAAID. In Case 1, the diagnosis of MAAID was reached within several months, since the genetic test was performed by Sanger sequencing of the *FAH* and *GSTZ1* genes. During this period, the patient was treated with Nitisinone and a tyrosine/phenylanine-restricted diet, which were discontinued when the definitive diagnosis was reached. In Case 1, NTBC and a tyrosine/phenylalanine-restricted diet were initiated at diagnosis because HT-1 could not initially be excluded. Before treatment, the NBS DBS showed elevated SA with normal tyrosine levels, and confirmatory UOA analysis revealed the presence of maleylacetoacetic acid. During treatment, SA levels on DBS decreased within the reference range, and circulating tyrosine transiently increased. Clinically, the patient maintained normal growth trajectories and showed no signs of hepatic or renal dysfunction. Following the definitive diagnosis of MAAID and discontinuation of treatment at 21 months, SA levels rose modestly (yet within the reference range), but there were no associated abnormalities in plasma tyrosine, liver transaminases, bilirubin, coagulation profile, or renal parameters, as well as growth and development at the last follow-up. The absence of clinical and biochemical deterioration after therapy withdrawal suggests that the intervention did not confer a measurable benefit in MAAID, at least during the follow-up period.

In Case 2, the final diagnosis was reached within a few weeks, making use of an NGS multigene panel. In Case 1, no treatment was started. Both infants remained asymptomatic during the follow-up period of 4 and 2 years, respectively. Our data are in line with previous sparse reports (Table 2), strengthening the observation that subjects with MAAID do not develop symptoms and arguing against the need to start any treatment. In agreement with Yang et al. [6], we suggest regular clinical, biochemical and imaging surveillance in MAAID.

Moreover, our data demonstrate the potential of a tyrosinemia-focused NGS gene panel in the differential diagnosis of metabolic overlapping disorders included in NBS panels. Indeed, the impact of MAAID on NBS is extremely relevant. Elevation of SA on DBS has long been considered pathognomonic of HT-1, thus leading to the activation of the proper clinical care pathway. On the other hand, a relatively high amount of false positive recalls is observed [18], raising significant clinical and organizational burden, and potentially resulting in unnecessary prescription of dietary and pharmacological treatment. Hence, timely identification of MAAID is paramount to avoid overload of care services, family stress, and unnecessary therapies. To this aim, different strategies can be pursued. One option could be to increase the NBS cut-offs for SA to avoid detection of MAAID. However, mild HT-1 cases may present with only mild SA elevation [19,20]. An alternative approach is to identify novel biomarkers. Urinary maleic acid levels appear to discriminate MAAID from HT-1, showing the potential to develop into a second-tier test [21]. In a recent study by van Vliet et al. [21], quantitative urine maleic acid (Q-uMA) measurement by LC-MS/MS successfully differentiated genetically confirmed MAAID cases from true HT-1 patients, with elevated levels detected in all MAAID cases and most false-positive newborn screening referrals, but in none of the HT-1 cases. These findings suggest that maleic acid could serve as a specific biomarker to reduce false-positive SA results and optimize post-screening diagnostic pathways. However, its application in routine newborn screening is not yet established.

Technical challenges remain, including assay standardization across laboratories, the need to validate analytical methods for dried blood spots, and the harmonization of reporting units. In addition, robust cut-off values must be defined, taking into account age-related reference ranges, possible dietary influences, and potential interferences from other organic acids. Large, multicentre validation studies are warranted to confirm diagnostic accuracy, assess the impact on screening performance metrics, and determine the feasibility and cost-effectiveness of integrating maleic acid measurement as a second-tier test in national newborn screening programs. The analysis of the dried blood spots is technically complex: measurement requires LC-MS/MS methods with specific optimizations and strict quality controls, as maleic acid is unstable and can undergo degradation or interference. Its concentration in dried blood spot samples is low, making a highly sensitive detection limit necessary and reducing the robustness of the test.

One additional strategy is the use of (first-/second-tier) genotyping combined with biochemical phenotyping as proposed for other genetic disorders [22]. Indeed, thorough molecular characterization is paramount to enable appropriate treatment decisions after positive NBS.

Furthermore, we identified for the first time the c.464_471delinsCTGGG and c.295G>A variants in the *GSZT1* gene and confirmed by in vivo analysis the pathogenicity of the c.68-12G>A variant. This latter variant had already been identified in compound heterozygosity with the c.259C>T variant in an asymptomatic patient diagnosed with MAAID [6]. However, in the study by Yang et al. [6], a functional analysis of the c.68-12G>A variant was not performed; therefore, its pathogenic role remained undefined. Our study demonstrates that this intronic variant has a deleterious effect on the splicing process, leading to a frameshift in the mRNA reading frame and likely resulting in the formation of a premature stop codon.

In summary, our findings expand the current clinical and molecular knowledge on MAAID, supporting the limited clinical relevance of this condition. Current evidence suggests that pharmacological treatment may not be necessary in the management of MAAID. Additional follow-up data are demanded to clarify the long-term prognosis of these individuals and define the optimal management strategy.

## Figures and Tables

**Figure 1 genes-16-01009-f001:**
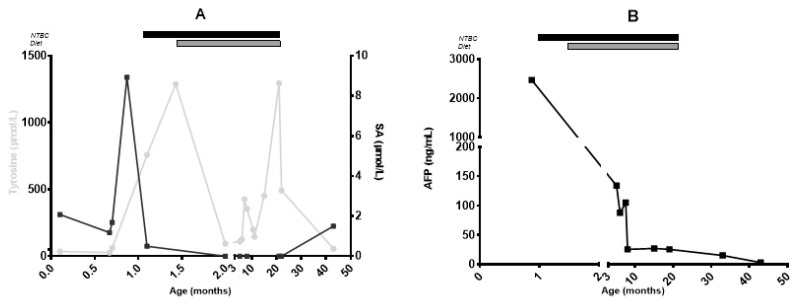
Tyrosine (grey line), Succinylacetone (SA, black line) (panel **A**), and alphafetoprotein (AFP, black line) (panel **B**) courses in case 1. NTBC was started at 1.0 months and discontinued at 22 months; diet was started at 1.4 months and discontinued at 22 months. Diet—tyrosine/phenylalanine-restricted diet; NTBC—Nitisinone. Reference ranges: Tyrosine: 22–108 µmol/L; Succinylacetone < 1.8 µmol/L. AFP reference range varies with age.

**Figure 2 genes-16-01009-f002:**
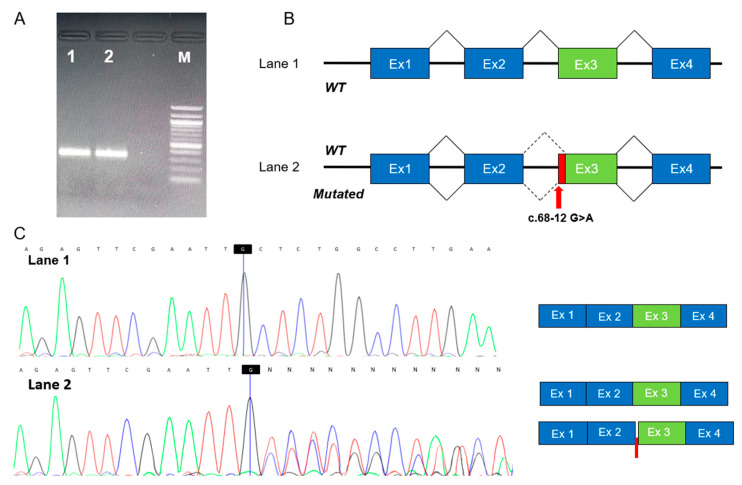
(**A**) Electrophoresis of the RT-PCR analysis of mRNA extracted from the peripheral blood. Lane 1: Negative control, Lane 2: Patient carrying the *GSTZ1*-c.68-12G>A mutation; M: Marker XIV (Roche Diagnostics). (**B**) Schematic representation (not to scale) of the splicing process in pre-mRNA without and with the *GSTZ1*-c.68-12G>A mutation. The red arrow represents the position of the mutation; the red box represents the insertion of ten nucleotides occurring in exon 3 of *GSTZ1*. (**C**) Electropherograms obtained from cDNA sequencing of the fragments shown in panel A (left) and a schematic representation of the corresponding splicing results (right). Insertion of ten nucleotides is represented by a red box. The blue line in the electropherograms indicates the end of exon 2 of *GSTZ1* cDNA.

**Table 1 genes-16-01009-t001:** Clinical and Biochemical Data of Patients (P1 and P2).

**PATIENT**				**Follow-Up (Main Points)**			
	**Biochemical/** **Clinical Data**	**NBS Recall**	**First** **Evaluation**	**I↑**	**II↓**	**III**	**Last** **Evaluation**
P1	Age (months)	0.1	0.7	0.9	1.1	33	43
	SA (µmol/L)	2.08	1.18	8.92	0.5	Undetectable	1.5
	Tyr (µmol/L)	36	27	N.A.	758	69	74
	AST (U/L)	-	72	49	70	30	45
	ALT (U/L)	-	28	31	59	18	49
	GGT (U/L)	-	160	150	196	18	24
	AFP (ng/mL)	-	2467	-	-	15.2	3.1
	SA urine µM/mM creatinine	-	1	36.5	-	-	5
	Liver Ultrasound	-	-	Regular size without focal lesions	-	Mild hepatomegaly (right lobe LD 130 mm) without focal lesions	-
	Weight (Kg/centile)	3.950 (75°)	4.800 (75°)	-	4.900 (75°)	16.800 (90°)	22 (>95°)
	Length (cm/centile)	-	58 (90–95°)	-	58 (90°)	99 (90°)	103(75°)
	Weight/Length (centile)	-	50°	-	25°	-	-
	Head Circumference (cm/centile)	-	38 (75°)	-	39 (75°)	-	-
	BMI (centile)	-	-	-	-	16.6 (50–75°)	21.1, (>95°)
P2							
	Age (months)	0.1	0.3	1	5.5	12	18
	SA (µmol/L)	2.24	1.48	0.48	-	-	0.49
	Tyr (µmol/L)	24.5	79	128	-	73	102
	AST (U/L)	-	92	39	55	47	--
	ALT (U/L)	-	21	20	27	22	--
	GGT (U/L)	-	64	101	11	10	--
	AFP (ng/mL)	-	-	-	-	-	8.30
	Weight (Kg/centile)	2.940	2.980 (25°)	3.900 (25–50°)	6.060 (10–25°)	7800 (10–25°)	9.320 (5–10°)
	Length (cm/centile)	51 (50–75°)	51 (50–75°)	53 (25–50°)	64 (25°)	73 (25–50°)	79 (25–50°)
	Weight/Length (centile)	5°	5°	25–50°	25°	5–10°	5–10°
	Head Circumference (cm/centile)	-	-	-	41.3 (10–25°)	-	45 (10–25°)
	Liver Ultrasound	-	-	Regular size without focal lesions	-	-	Regular size without focal lesions
	Heart Ultrasound	-	-	Ostium secundum atrial septal defect with moderate shunt	-	normal	-

AFP: Alpha-Fetoprotein (r.v.316–6310); ALT: Alanine Aminotransferase (r.v.10–50); AST: Aspartate Aminotransferase (r.v.10–50); GGT: Gamma-glutamyl transferase (r.v.10–71); SA: Succinylacetone (r.v. <1.8, blood, undetectable, urine); Tyr: Tyrosine (r.v. 22–108); N.A.: Not analised. Blood counts, electrolytes, kidney function, and inflammatory and coagulation indices were in the reference ranges for both patients. The upward arrow pointing downward indicates the start of NTBC. The downward arrow pointing up indicates the diet and NTBC stopping. Detailed data on Succinylacetone and Tyrosine trends in Case 1 are presented in Figure 1.

**Table 2 genes-16-01009-t002:** Individuals with MAAID described in the literature for whom clinical information is available. N.A.—Not Available * Family segregation analysis not available.

Case	Age at Last Follow-Up (Years)	Sex	Allele 1	Allele 2	Clinical Features	Treatment	Reference
1	13.3	M	c.259C>T (p.Arg87Ter)	c.68-12G>A	Asymptomatic	None	[6]
2	10.3	F	c.449C>T (p.Ala150Val)	c.449C>T (p.Ala150Val)	Asymptomatic	None	[6]
3	10	M	c.449C>T (p.Ala150Val)	c.449C>T (p.Ala150Val)	Asymptomatic	None	[6]
4 *	2.5	F	c.449C>T (p.Ala150Val)	c.449C>T (p.Ala150Val)	Asymptomatic	None	[6]
5 *	1.6	F	c.449C>T (p.Ala150Val)	c.449C>T (p.Ala150Val)	Asymptomatic	None	[6]
6	1.4	M	c.295G>A (p.Val99Met	N.A.	Asymptomatic	None	[6]
7	1.3	M	c.136−2A>G	c.136−2A>G	Microcephaly, short stature	None	[7]
8	32	M	c.136−2A>G	c.136−2A>G	Asymptomatic	None	[7]
9	3.5	M	c.68-12G>A	c.464_471delinsCTGGG (p.Val155_Asp157delinsAlaGly)	Obesity	First NTBC and protein restriction, then discontinued	Current report
10	1	F	c.68-12G>A	c.295G>A (p.Val99Met)	Asymptomatic	None	Current report

## Data Availability

All data are available from the corresponding author upon request.

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
