# Peer review of "Variants in GSTZ1 Gene Underlying Maleylacetoacetate Isomerase Deficiency: Characterization of Two New Individuals and Literature Review"

_genes, 2025, doi:10.3390/genes16091009_

Round 1
Reviewer 1 Report
Comments and Suggestions for Authors
This manuscript reports two new cases of MAAID with novel GSTZ1 variants and includes functional validation of a splicing variant, which adds valuable information to the limited literature on this rare metabolic disorder. The genetic and biochemical analysis is well presented, and the discussion addresses the diagnostic implications in the context of newborn screening.
- The introduction focuses heavily on HT-1 but provides little background on MAAID. Since the manuscript centers on MAAID, the authors should clearly explain why these cases are worth reporting, how MAAID differs from other tyrosine metabolism disorders, and what clinical or diagnostic value this report adds.
- Although the ethics approval and consent statement is included at the end of the manuscript, it is recommended to move this information to the Methods section for improved clarity and compliance with standard reporting practices.
- As this appears to be the only reported MAAID case to have undergone treatment, a more explicit discussion comparing clinical and biochemical data before and after treatment would help clarify whether intervention was necessary.
- The potential of maleic acid as a second-tier biomarker is intriguing, but it may benefit from a brief discussion on its current validation status or technical challenges.
Author Response
This manuscript reports two new cases of MAAID with novel GSTZ1 variants and includes functional validation of a splicing variant, which adds valuable information to the limited literature on this rare metabolic disorder. The genetic and biochemical analysis is well presented, and the discussion addresses the diagnostic implications in the context of newborn screening.
1.The introduction focuses heavily on HT-1 but provides little background on MAAID. Since the manuscript centers on MAAID, the authors should clearly explain why these cases are worth reporting, how MAAID differs from other tyrosine metabolism disorders, and what clinical or diagnostic value this report adds.
We thank the reviewer for his/her suggestion. The Introduction has been expanded and revised according to his/her recommendations (see Introduction section, pag. 2, lines 50-80, of the modified manuscript).
2.Although the ethics approval and consent statement is included at the end of the manuscript, it is recommended to move this information to the Methods section for improved clarity and compliance with standard reporting practices.
Taking into account the Reviewer’s point we moved the ethics approval and consent statement to the Methods section (see pag. 4, lines 136-138, of the revised manuscript).
3.As this appears to be the only reported MAAID case to have undergone treatment, a more explicit discussion comparing clinical and biochemical data before and after treatment would help clarify whether intervention was necessary.
According to reviewer suggestions, we compared clinical and biochemical data before and after the treatment and discussed the appropriateness of the treatment (see pag. 9, lines 256-269, of the revised manuscript).
4.The potential of maleic acid as a second-tier biomarker is intriguing, but it may benefit from a brief discussion on its current validation status or technical challenges.
The preliminary findings shown in our paper are promising, however maleic acid measurement is not yet part of routine newborn screening workflows. Current evidence derives from a limited number of cases, and no large-scale validation studies have been published to date. Moreover, technical challenges remain, including assay standardization, establishing reliable cut-offs, and evaluating potential interferences from other metabolites. Future multicentre studies will be essential to confirm diagnostic accuracy and to assess feasibility for integration as a second-tier test in clinical practice. We discussed these issues (see pag. 9-10, lines 287-307, of the revised manuscript).
Reviewer 2 Report
Comments and Suggestions for Authors
The manuscript by Barretta et al. reports two cases of maleylacetoacetate isomerase deficiency (MAAID). Both cases showed elevated succinylacetone (SA) and normal tyrosine levels on newborn screening: the first intermittently, and the second with no SA elevation on the confirmatory test. The authors characterize the patients at the clinical, biochemical, and genetic levels and provide in silico and in vivo prediction of the -12 variant.
In the discussion, the authors highlight that SA elevation occurs not only in tyrosinemia type 1 but also in MAAID. They provide a table comparing the phenotypes and discuss diagnostic methods, including the use of an NGS panel. The text is concise, with the most important aspects clearly presented. The figures are representative, and the most important references are included.
This is a short but clear case report. Given the limited information available on MAAID, it is worth publishing.
Comments:
-
It would be better to reclassify the manuscript as a case report.
-
The c.68-12G>A variant has already been reported once as causative. This should be commented on in the discussion.
Author Response
The manuscript by Barretta et al. reports two cases of maleylacetoacetate isomerase deficiency (MAAID). Both cases showed elevated succinylacetone (SA) and normal tyrosine levels on newborn screening: the first intermittently, and the second with no SA elevation on the confirmatory test. The authors characterize the patients at the clinical, biochemical, and genetic levels and provide in silico and in vivo prediction of the -12 variant.
In the discussion, the authors highlight that SA elevation occurs not only in tyrosinemia type 1 but also in MAAID. They provide a table comparing the phenotypes and discuss diagnostic methods, including the use of an NGS panel. The text is concise, with the most important aspects clearly presented. The figures are representative, and the most important references are included.
This is a short but clear case report. Given the limited information available on MAAID, it is worth publishing.
Comments:
1.It would be better to reclassify the manuscript as a case report.
According to reviewer suggestion we reclassified the manuscript as Case Report
2.The c.68-12G>A variant has already been reported once as causative. This should be commented on in the discussion.
The c.68-12G>A variant had already been identified in compound heterozygosity with the c.259C>T variant in an asymptomatic patient diagnosed with MAAID by Yang et al. However, the authors did not perform a functional analysis of the c.68-12G>A variant; therefore, its pathogenic role remained undefined. Our study demonstrates that this intronic variant has a deleterious effect on the splicing process, leading to a frameshift in the mRNA reading frame and likely resulting in the formation of a premature stop codon. We reported these observations in the Discussion (see pag. 10, lines 314-320, of the revised manuscript).